

# The efficacy of aerosol-cloud-radiative perturbations from near-surface emissions in deep open-cell stratocumulus

Anna Possner[1], Hailong Wang[2], Robert Wood[3], Ken Caldeira[2], and Thomas P. Ackerman[3,4]

[1]Carnegie Institution for Science
[2]Pacific Northwest National Laboratory
[3]Department of Atmospheric Sciences, University of Washington
[4]Joint Institute for the Study of the Atmosphere and Ocean, University of Washington

*Correspondence to:* Anna Possner (apossner@carnegiescience.edu)

**Abstract.** Aerosol-cloud-radiative effects are determined and quantified in simulations of deep open-cell stratocumuli observed during the VOCALS-REx campaign off the West coast of Chile. The cloud deck forms in a 1.5 km deep boundary layer with cell sizes reaching 50 km in diameter. Global data bases of ship tracks suggest that these linear structures are seldom found in boundary layers this deep. Here, we quantify the changes in cloud-radiative properties to a continuous aerosol point source

moving along a fixed emission line releasing $10^{17}$ particles per second. We show that a spatially coherent cloud perturbation is not evident along the emission line. Yet, our models simulates an increase in domain-mean all-sky albedo of 0.05 corresponding to a diurnally-averaged cloud-radiative effect of $20 \, \mathrm{W \, m^{-2}}$ given the annual mean solar insolation at the VOCALS-REx site. Therefore, marked changes in cloud-radiative properties in precipitating deep open cells may be driven by anthropogenic near-surface aerosol perturbations such as ships.

Furthermore, we demonstrate that these changes in cloud-radiative properties are masked by the natural variability within the organised cloud field. A clear detection and attribution of cloud-radiative effects to a perturbation in aerosol concentrations becomes possible when sub-filtering of the cloud field is applied utilising the spatio-temporal distribution of the aerosol perturbation. Therefore, this work has implications for the detection and attribution of effective cloud-radiative forcing in marine stratocumuli, which constitutes one of the major physical uncertainties within the climate system. Our results suggest that ships

may sometimes have a substantial radiative effect on marine clouds and albedo even when ship tracks are not readily visible.

## 1 Introduction

Aerosol-cloud interactions (aci) in low-level clouds, which span just over a fifth of the Earth's ocean surface (Wood, 2012), contribute the largest uncertainty to estimates of global mean effective radiative forcing (ERF) of aerosols (Myhre et al., 2013).

Current estimates of $\mathrm{ERF}_{aci}$ range from $-1.2 \, \mathrm{W \, m^{-2}}$, which constitutes a strong global cooling partially offsetting the effects of warming due to anthropogenic greenhouse-gas emissions, to $0 \, \mathrm{W \, m^{-2}}$, rendering these effects negligible at the global scale (Boucher et al., 2013).

Reducing this uncertainty substantially using satellite retrievals and global climate models (GCMs) remains challenging due to



issues of collocation in retrievals of aerosol and cloud properties from space (Koren et al., 2007; Charlson et al., 2007) and the inadequate representation of small-scale dynamical processes contributing to the cloud response in coarse-scale models (Nam et al., 2012; Schneider et al., 2017). Valuable insights on the involved processes and plausible ranges of cloud-radiative perturbations through aerosols have been obtained by the study of ship tracks. These *anomalous cloud lines* (Conover, 1966) are

a phenomenon associated with a characteristic spatial structure which occurs in low-level stratocumulus clouds. The changed cloud-radiative properties within these tracks can be attributed to localised aerosol perturbations (Durkee et al., 2000b, a).

Data bases obtained from satellite retrievals (Coakley and Walsh, 2002; Christensen and Stephens, 2012; Chen et al., 2015) as well as high-resolution modelling studies (Wang and Feingold, 2009; Wang et al., 2011; Berner et al., 2015) show that the net cloud-radiative effect (CRE) in individual ship tracks does not only depend on cloud droplet number increases and size

decreases, which occur in almost all cases (Chen et al., 2012), but also on induced changes in cloud morphology, cloud fraction and liquid water path ($LWP$). Cloud albedo ($A_{cld}$) may not always increase with increased levels of pollution, but may also decrease (Christensen and Stephens, 2012; Berner et al., 2015). Furthermore, localised gradients in aerosol concentration have been shown to induce self-sustaining mesoscale circulations (Chen et al., 2015; Wang et al., 2011) by the local suppression of precipitation in the polluted cloud and the convergence of cold pools transporting moisture into the polluted cloud from the

surrounding precipitating clouds. In the global mean, $LWP$ increases in ship tracks forming within the precipitating cloud regime irrespective of above-cloud moisture content (Toll et al., 2017).

However, ship tracks are rare in comparison to the number of ocean-going ships criss-crossing the world's oceans. Merely 1924 ship tracks were detected over two years worldwide (Campmany et al., 2009), while the total ocean-going fleet consists of over 50,000 ships exceeding 500 gross tons in weight (Agency, 2014). Understanding what constrains their occurrence in terms of

background pollution, boundary layer dynamics and large-scale stability, may facilitate constraining regimes and magnitudes of global effective radiative forcing estimates. In particular, studies of ship tracks in high-resolution models and satellite retrievals have been mostly limited to extremely shallow boundary layers ranging in depth from 300 m to 600 m (Christensen and Stephens, 2012; Berner et al., 2015; Chen et al., 2015). Ship track formation within one slightly deeper boundary layer of 800 m was investigated in high-resolution simulations by Wang and Feingold (2009) and Wang et al. (2011). Indeed ship tracks

are very rarely detected in satellite retrievals of boundary layers deeper than 800 m (Durkee et al., 2000b; Toll et al., 2017). Yet, over 70% of stratocumulus clouds are found in deeper boundary layers (Muhlbauer et al., 2014).

The potential for albedo changes is particularly high in the open-cell, and disorganised stratocumulus regimes, which occur more frequently in the sub-tropics, than in the closed cells regime (Muhlbauer et al., 2014). Both of these regimes are characterised by shallow convective cloud structures detraining laterally at cloud top. The detrained cloud sheets spanning the regions

between the convective structures are optically thin (cloud optical thickness $\tau < 3$), are often associated with low droplet number concentrations ($N_d \sim 5$), and may contribute substantially to the overall cloud fraction (Wood et al., 2018). Thus, their albedo is highly susceptible to aerosol perturbations from the perspective of Platnick and Twomey's albedo susceptibility definition. Yet, the efficacy of aerosol-cloud-radiative interactions within these detrained cloud segments remains unclear. In general, the processes governing aerosol-cloud interactions in deep stratocumulus boundary layers remain weakly constrained

with only few process-level studies (Wang et al., 2010; Kazil et al., 2011; Wood et al., 2011b; Zuidema et al., 2016) quantifying





effects on cloud characteristics through aerosol pollution.

Within this study we quantify changes in cloud-radiative properties due to aerosol perturbations in deep (boundary layer depth of $\sim 1.5\,\mathrm{km}$) open-cell stratocumulus clouds and discuss dominant mechanisms which constrain the cloud-albedo response.

## 2 Methodology

### 2.1 Case description

This study is based on a well-documented case of open-cell stratocumulus clouds embedded within a $\sim 1.5\,\mathrm{km}$ deep boundary layer, which was observed during research flight 6 of the VAMOS Ocean-Cloud-Atmosphere-Land Study Regional Experiment (VOCALS-REx) campaign. Detailed information on the particular case and measurement techniques can be found in Wood et al. (2011a) and Wood et al. (2011b) respectively. Here an overview of the two cloud regimes and their characteristics relevant to this study is given.

**Table 1.** Time-mean quantities of liquid water path ($LWP$), surface precipitation ($\mathrm{R_{sfc}}$), cloud base precipitation ($\mathrm{R_{cb}}$), cloud fraction (CF), sub-cloud mean boundary layer aerosol concentration ($N_{a\_sub}$) and cloud mean droplet number concentration ($N_{d\_top}$). The first row containing data shows the observations of the open-cell stratocumulus deck obtained during research flight RF06 of the VOCALS-Rex campaign on October, $28^{th}$, 2008 between $08{:}00-13{:}30$ UTC. The numerical results, shown in the last two rows, were averaged over the identical time periods over both simulated days. Domain mean values were computed for all variables other than the surface precipitation field which was averaged only over values exceeding 0.1 mm/day (consistent with observations). Numbers in brackets denote the interquartile range of each variable. Numerical results are shown for the control simulation (*ctrl*) and the aerosol-perturbed simulation (*ship*). Further details on simulations can be obtained in text.

| | | | $08{:}00-13{:}30$ UTC averages | | | |
|---|---|---|---|---|---|---|
| Sim/obs | LWP [g m$^{-2}$] | $\mathrm{R}_{sfc}$ [mm day$^{-1}$] | $\mathrm{R}_{cb}$ [mm day$^{-1}$] | CF [%] | $\mathrm{N}_{a\_sub}$ [cm$^{-3}$] | $\mathrm{N}_{d\_top}$ [cm$^{-3}$] |
| RF06 | 141 | 1 | $4-5$ | 56 | 30 | 10 |
| *ctrl* | 75 [6, 70] | 4.2 [0.2, 2.4] | 9.1 [0.1, 2.9] | 50 | 34 [32, 37] | 8 [3, 11] |
| *ship* | 82 [10, 81] | 4.1 [0.2, 2.2] | 8.6 [0.1, 2.9] | 57 | 45 [32, 42] | 12 [3, 13] |

The cloud regime was sampled during the early morning hours (03 am to 08:30 am local time) on October $28^{th}$ in 2008. A summary of numerous cloud and boundary layer properties measured during the campaign is given in Table 1. The characteristic cell size was found to be between $30-40\,\mathrm{km}$, which is detected frequently in Southeast Pacific stratocumulus clouds (Wood and Hartmann, 2006). A cloud fraction of 56 % was measured in the open-cell regime, which is consistent with the observed high level of detrained cloudy air masses, which spread from the updraft cores into inner regions of the cell. Furthermore, a





cloud cover of this extent is typical for marine open-cell stratocumulus (Muhlbauer et al., 2014; Terai et al., 2014). The open-cell clouds coincided with moister sub-cloud layer air masses, as compared to the neighbouring closed-cell regime, and were characterised by low sub-cloud layer aerosol concentrations ($30 \, cm^{-3}$). A strong vertical gradient in aerosol concentration was observed within the open cells near cloud base where concentrations decreased rapidly. A strong horizontal

gradient in cloud-top droplet number concentration ($N_{d\_top}$) was observed (Wood et al., 2011a) between the updraft cores ($N_{d\_top} \sim 30 \, cm^{-3}$) and the detrained cloud filaments ($N_{d\_top} \sim 1 - 10 \, cm^{-3}$). Substantial rates of drizzle were observed at cloud base. Yet, over $50 \, \%$ evaporated before reaching the surface.

## 2.2 Simulation Setup

Two simulations were performed with the weather research forecast (WRF) model at the convection-resolving scale with a horizontal grid resolution of $300 \times 300 \, m^2$, a vertical resolution of $30 \, m$ and a time step of $3 \, s$ following Wang et al. (2010). The idealised simulations with periodic boundary conditions at the domain edges were initialised with meteorological profiles obtained during research flight 6 of the VOCALS-REx field campaign (Wang et al., 2011; Wood et al., 2011a). A brief overview of the research flight is given in the previous section 2.1. Given the large characteristic spatial scales of the cellular organisation

of the cloud field with cell sizes ranging from $30 \, km$ to $40 \, km$, simulations were performed on a large domain of $180 \times 180 \, km^2$. The domain was centered on $78°W$ and $15°S$, which is off the west coast of Chile. The model top was specified at an altitude of $2 \, km$, which is $600 \, m$ above the boundary layer top. Above this height a standard clear-sky atmosphere profile is assumed for the computation of the radiative fluxes until the top of atmosphere.

Both simulations were run for $48 \, h$ with a fixed divergence rate of $1.67 \times 10^{-6} \, s^{-1}$, which was estimated from QuickSAT

surface winds, and prescribed surface fluxes. Surface latent heat and sensible heat fluxes were specified according to field measurements as $120 \, W \, m^{-2}$ and $15 \, W \, m^{-2}$ (defined as +ve upward) respectively. The surface pressure was specified as $1018 \, hPa$. For simplicity mean advective tendencies in the wind field were removed from the soundings.

The simulations were performed with the two-moment Morrison et al. (2009) microphysics scheme with a prognostic treatment of number and mass concentrations for cloud water and rain. The exponents of the cloud liquid water content and $N_d$ in the

Khairoutdinov and Kogan (2000) autoconversion rate were adjusted to values obtained from the VOCALS-REx field campaign as 3.19 (cloud water exponent) and $-1.49$ ($N_d$ exponent) respectively. These exponents were obtained for the VOCALS-REx field data using the approach described in Wood (2005). Precipitation formation was artificially suppressed in the first $2 \, h$ of simulation to facilitate a thermodynamic adjustment to the initialisation sounding before including moisture sinks. Cloud condensation nuclei (CCN) were treated as in Wang et al. (2011) with a prognostic log-normal sea-salt mode centered at a

mean diameter of $500 \, nm$ and variance of 1.5. Aerosols were advected according to grid-scale and subgrid-scale transport tendencies and aerosol-cloud interactions were included by removing aerosols upon activation, which was treated as in Kravitz et al. (2014). The release of aerosol upon complete evaporation of cloud droplets and rain drops was also simulated. A surface sea-salt emission flux of $20 \, m^{-2} \, h^{-1}$ was specified in line with estimates from previous simulations (Wang et al., 2010; Kazil et al., 2011).



In addition to the control simulation, from here on named *ctrl* simulation, an aerosol-perturbation experiment was designed. The simulation named *ship* simulation, followed the setup of Wang et al. (2011) for direct comparison between the deep boundary layer case and shallow boundary layer case in terms of aerosol-cloud-radiative perturbations. A ship moving at $5\,\mathrm{m\,s^{-1}}$ through the center of the domain was allowed to continuously emit sea-salt at a rate of $10^{17}\,\mathrm{s^{-1}}$ and a mean dry radius of $300\,\mathrm{nm}$. This

5  flux was chosen to match emissions within a previously studied case within a shallow open-cell regime (Wang et al., 2011) (see section 3.2.1 for an in-depth discussion). Furthermore, these emissions were consistent with estimates proposed by Salter et al. (2008) for marine cloud brightening applications. The CAM radiation scheme was used in the simulations and $A_{cld}$ was estimated as $A_{cld} = \tau/(\tau + 6.8)$, where $\tau$ denotes the cloud optical depth which in turn was diagnosed as:

10  $$\tau = \int_{z=0}^{h} \frac{3q_l}{2\rho_w R_{eff}} dz, \qquad \rho_w = \text{density of water}, q_l = \text{liquid water content}, R_{eff} = \text{effective cloud droplet radius.} \tag{1}$$

The cloud base precipitation rate ($R_{cb}$) was computed as the mean precipitation flux across the lowest third of the cloud vertical extent, which is consistent with its estimation from observations.



## 3 Results

### 3.1 Evaluation of open-cell characteristics

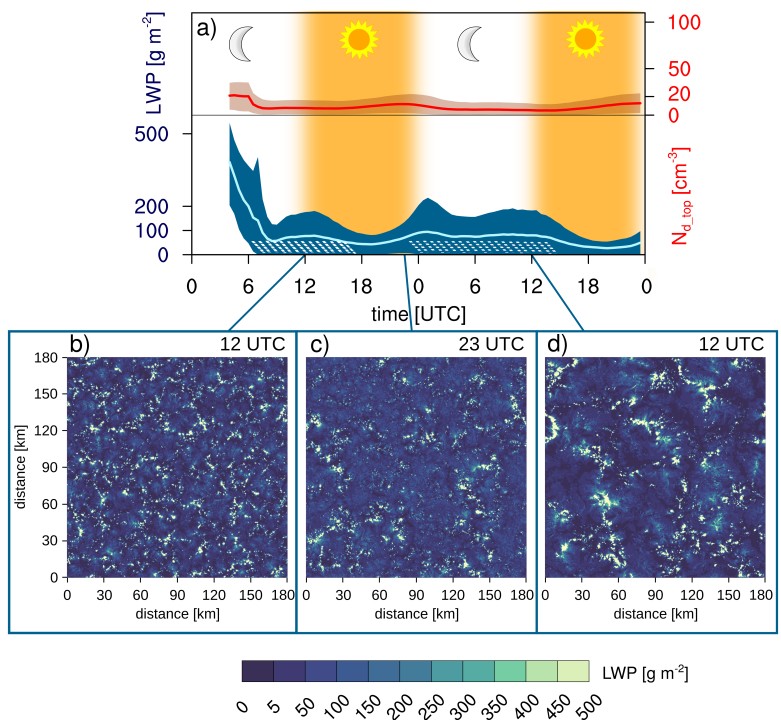

**Figure 1.** (a) Time series of domain-mean liquid water path, $LWP$ (green), and cloud-top droplet number concentration, $N_{d\_top}$ (red), for the *ctrl* simulation. Shading (blue for $LWP$, black for $N_{d\_top}$) denotes interdecile percentile range. Snapshots of $LWP$ are shown in (b) after initial organised structures developed, (c) after the solar maximum and (d) for the second day organised state.

In order to assess the radiative effect of concentrated and localised aerosol pollution on deep open-cell clouds, the simulations need to demonstrate sufficient skill in capturing the characteristics and dynamics of the open-cell regime. Following initialisation, an unorganised stratiform cloud deck formed in the *ctrl* simulation. Initial organised structures appeared 6 h after initialisation following the onset of precipitation (Fig. 1). Following the second night, realistic length scales of organisation (see section 2.1) were simulated.

The diurnal evolution of $LWP$ and $N_{d\_top}$ is shown in Fig. 1. Periods, when $R_{cb}$ exceeds $3\,\mathrm{mm\,day^{-1}}$ are marked in white. The simulation showed a pronounced diurnal cycle during both days in $LWP$ and $R_{cb}$. As in Wang et al. (2010), solar heating was found to break up the cell-walls, which led to a reduction of $LWP$ in the upper percentiles, a reduction in cloud-base precipitation rates to $R_{cb} < 2\,\mathrm{mm\,day^{-1}}$, and consequently the loss of cloud-field organisation in the late afternoon. During the




night the cloud deck recovered and organisation was re-established.

The *ctrl* simulation was characterised by a well-mixed cloud layer and stably stratified sub-cloud layer (see Fig. S1), which is characteristic for deep boundary layers. This structure developed rapidly following initialisation from the well-mixed state. Within the first 3.5 h the boundary layer deepened by 180 m before stabilising at 1.5 km and the sub-cloud layer became strati-

fied. A mean $R_{cb}$ of 9.1 mm day$^{-1}$ (Table 1) was simulated in the early morning hours of the VOCALS-REx field campaign. Although simulated mean $R_{cb}$ was within the spread of observed precipitation rates, it was roughly twice as high as the mean $R_{cb}$ rates inferred from observations (Table 1). Meanwhile, the mean $LWP$ was underestimated by a factor 2 in the open-cell regime, which is consistent with an over-estimation in precipitation. However, the simulated cumulative precipitation distribution shown in the supporting material (Fig. S2a) showed that the overall distribution of $R_{cb}$ was well captured in the *ctrl*

simulation and that the bias in the mean originates from the slight overestimation of intense precipitation events exceeding 20 mm day$^{-1}$. These events are likely to be found within the walls of the open-cells, which are characterised by strong updrafts (Fig. S2b).

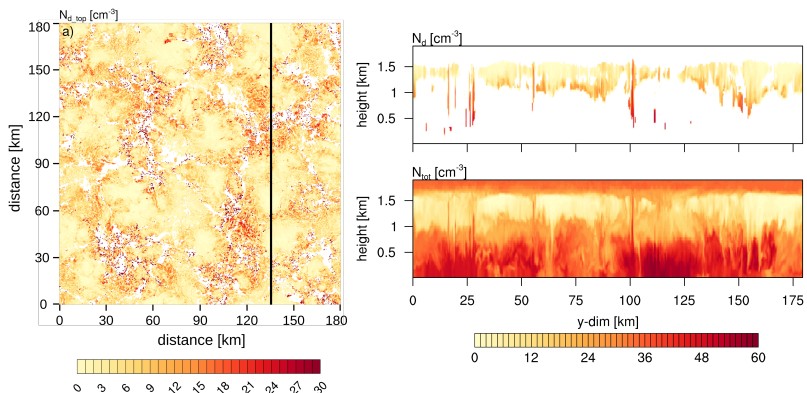

**Figure 2.** (a) instantaneous cloud-top droplet number concentration, $N_{d\_top}$ (corresponding snap shot to $LWP$ field shown in Fig. 1d). Black line denotes location of cross-section of (b) cloud droplet number concentration ($N_d$) and (c) total number concentration ($N_{tot} = N_a + N_d$, where $N_a$ denotes the aerosol number concentration).

The microphysical quantities, such as the mean sub-cloud layer aerosol concentration ($N_{a\_sub} = 34$ cm$^{-3}$) and $N_{d\_top} =$

8 cm$^{-3}$ were in good agreement with the observations. In the simulations the aerosol particles are lifted into the cloud layer within the cell-walls, where they activated and $N_d$ was relatively high. Cloud filaments, many of which are quite optically thin, were detrained horizontally, and are characterized characterised by low $N_{d\_top}$ (Fig. 2a) due to the efficient removal through precipitation. While cell-wall $N_d$ may reach up to 40 cm$^{-3}$ in the *ctrl* simulation (Fig. 2b), characteristic $N_d$ in detrained cloud filaments, sometimes reffered to as "veil clouds" (Wood et al., 2018), were as low as 2–3 cm$^{-3}$. The efficient removal of

aerosol particles through cloud processing combined with the stable stratification in the sub-cloud layer induced strong vertical gradients in the combined particle number concentration $N_{tot}$ defined as $N_{tot} = N_a + N_d$, where $N_a$ denotes the aerosol num-





ber concentration. Sub-cloud layer $N_{tot}$ ranged between $30 - 60\,\text{cm}^{-3}$, while values below $10\,\text{cm}^{-3}$ above cloud base height were simulated frequently (Fig.2c).

In summary, despite remaining biases in the domain mean $LWP$ and $R_{cb}$, the simulation overall captured a realistic evolution of the open-cell cloud deck with a pronounced diurnal cycle. Since the overall cloud-cell statistics (Table. 1) and the horizontal

cloud cover are consistent with observations, it gives us confidence that the underlying cloud dynamics were captured in the *ctrl* simulation. Regions of detrained cloud spanned 36% of the domain and were characterised by low $LWP$ and low $N_{d\_top}$, which makes them particularly susceptible to aerosol-induced cloud-radiative perturbations. Yet, any near-surface source of pollution will predominantly be transported into the cloud layer through the cell walls given the pronounced vertical stratification in the sub-cloud layer, where wet aerosol removal processes are efficient. It therefore remains to be seen whether

substantial changes in cloud-radiative properties can be induced by near-surface aerosol perturbations.

## 3.2 Efficacy of aerosol perturbation

**Table 2.** Same as Table 1 but for mean values of the last 24 h period. The following additional variables were added to the table: cloud albedo ($A_{cld}$), all-sky albedo$= CF * A_{cld} + (1 - CF) * A_{clr}$ and cloud-top effective cloud droplet radius ($R_{eff\_top}$). $A_{clr}$ denotes the clear-sky albedo which was determined as $A_{clr} = 0.06$ in both simulations.

| | | | | 2nd day mean | | | | | |
|---|---|---|---|---|---|---|---|---|---|
| Sim/obs | $LWP$ [g m$^{-2}$] | $R_{sfc}$ [mm day$^{-1}$] | $R_{cb}$ [mm day$^{-1}$] | $A_{cld}$ | Albedo | $CF$ [%] | $N_{a\_sub}$ [cm$^{-3}$] | $N_{d\_top}$ [cm$^{-3}$] | $R_{eff\_top}$ |
| *ctrl* | 64 [2, 57] | 4.6 [0.2, 2.4] | 8.4 [0.1, 2.3] | 0.18 [<0.01, 0.30] | 0.22 | 44 | 36 [32, 39] | 9 [3, 12] | 21 [17, 26] |
| detrained | 92 [38, 101] | 4.5 [0.2, 2.6] | 9.2 [0.3, 4.6] | 0.37 [0.28, 0.43] | 0.13 | 36 | – | 8 [3, 12] | 23 [19, 27] |
| wall | 331 [71, 449] | 10.0 [0.4, 7.8] | 33.7 [0.4, 18.5] | 0.61 [0.45, 0.77] | 0.05 | 8 | – | 13 [4, 18] | 21 [17, 25] |
| *ship* | 75 [10, 73] | 4.5 [0.2, 2.1] | 7.7 [0.1, 2.2] | 0.26 [0.07, 0.41] | 0.27 | 58 | 50 [33, 50] | 14 [4, 17] | 21 [16, 26] |
| *ship*-seeded | 87 [18, 88] | 4.8 [0.2, 2.2] | 7.6 [0.1, 2.0] | 0.33 [0.16, 0.48] | 0.35 | 72 | 77 [47, 81] | 24 [6, 31] | 18 [14, 22] |
| detrained | 87 [38, 99] | 4.1 [0.2, 2.1] | 5.9 [0.1, 2.4] | 0.43 [0.32, 0.52] | 0.25 | 59 | – | 25 [7, 32] | 19 [14, 22] |
| wall | 254 [54, 307] | 9.6 [0.3, 6.8] | 24.6 [0.2, 8.1] | 0.59 [0.43, 0.75] | 0.08 | 13 | – | 36 [10, 46] | 18 [14, 21] |
| *ship*-unseeded | 67 [5, 67] | 4.3 [0.2, 2.1] | 7.7 [0.1, 2.3] | 0.23 [0.04, 0.36] | 0.24 | 51 | 35 [31, 38] | 9 [3, 11] | 22 [18, 27] |
| detrained | 92 [41, 104] | 4.1 [0.2, 2.2] | 8.1 [0.4, 4.1] | 0.37 [0.29, 0.44] | 0.16 | 43 | – | 8 [3, 11] | 24 [19, 28] |
| wall | 299 [68, 414] | 9.6 [0.3, 7.1] | 30.2 [0.4, 15.2] | 0.57 [0.41, 0.73] | 0.05 | 8 | – | 12 [4, 17] | 22 [18, 26] |

The sea-salt perturbed simulation displayed a spatially constrained aerosol plume meandering around the emission line (Fig. 3a). The highest values of $N_{a\_sub}$ exceeding $1000\,\text{cm}^{-3}$ were found within a narrow plume up to $60\,\text{km}$ in the wake of the point

source. Overall, the aerosol perturbation remained spatially constrained within the boundary layer in a region of $\pm 30\,\text{km}$ around the emission line. This strip of the domain (spanning $60 - 120\,\text{km}$ in the y-direction) is characterised by increased levels of $N_{a\_sub}$ and will be from here on referred to as "seeded", whereas the domain outside this region will be referred to as "unseeded".



Inside the seeded region the emitted aerosol were predominantly transported into the cloud within the updrafts of the cell walls (Fig. S3). Despite efficient wet-removal processes within the cell walls, the largest absolute changes in $N_d$ were as large as $600\,\mathrm{cm}^{-3}$. At cloud-top, increases in $N_{d\_top}$ of up to $150\,\mathrm{cm}^{-3}$ were found (Fig. 3b). From the cell walls, the increased levels of $N_d$ persisted to the detrained cloud regions (Fig. S3), where the largest relative increase in $N_{d\_top}$ were found. On average

5    $N_{d\_top}$ increased by $177\,\%$ within the cell walls and by $213\,\%$ within the stratified detrained cloud (Table 2). In this analysis cell-walls were diagnosed as cloud-covered regions with updraft speeds exceeding $0.5\,\mathrm{m\,s}^{-1}$. All remaining, non-wall cloudy grid points were classified as "detrained cloud".



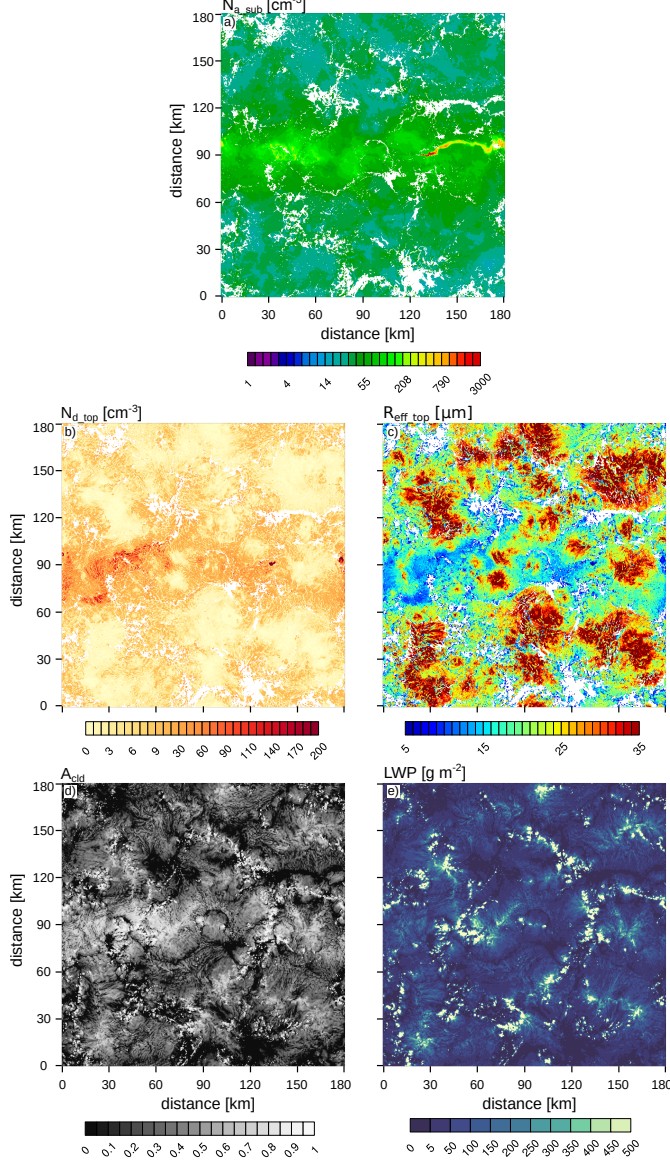

**Figure 3.** Snap shots of a) vertically averaged sub-cloud layer aerosol concentration ($N_{a\_sub}$), b) cloud-top cloud droplet number concentration ($N_{d\_top}$), c) cloud-top mean effective cloud droplet radius ($R_{eff\_top}$), d) cloud albedo ($A_{cld}$) and e) liquid water path ($LWP$). Instantaneous fields are shown at 12 UTC for the *ship* simulation. Fields for the *ctrl* simulation are shown in the supporting material (Fig. S4).

The largest decreases in the cloud droplet effective radius at cloud top ($R_{eff\_top}$) were found to coincide with regions of large increases in $N_{d\_top}$ (Fig. 3c and Fig. 3b respectively). $R_{eff\_top}$ may be reduced by up to $10\,\mu$m locally. The largest decreases in $R_{eff\_top}$ were found within the vicinity of strong updrafts. Here, many aerosols were carried into the cloud layer and were activated. Going radially outward from the updraft cores, efficient in-cloud scavenging led to a reduction in $N_{d\_top}$ and an





increase in $R_{eff\_top}$. Averaged over the seeded domain, a mean reduction in $R_{eff\_top}$ between $3-4\,\mu$m was simulated in the cell walls and detrained cloud regions (Table 2).

Averaged over the domain, the changes in cloud-microphysical properties led to an increase in domain-averaged $LWP$ within the seeded (36 %) and unseeded regions (5 %). Yet, lower mean-values of in-cloud $LWP$ were found within the detrained

cloud and cell-wall regimes in both domains of the *ship* simulation (Table 2). $R_{cb}$ was found to decrease by $0.7\,\mathrm{mm\,day}^{-1}$, while surface precipitation rates were largely unaffected by the aerosol perturbation (Table 2).

Due to the reduction in $R_{cb}$, more cloud water was retained within the updraft and detrained horizontally into the stratified cloud filaments, which penetrated deeper into the open cells. The increase in areal extent of the detrained cloud sheets was accompanied by a shift (Fig. S5) in in-cloud $LWP$ distribution towards lower $LWP$ between $50-150\,\mathrm{g\,m}^{-2}$. Therefore, the

10 increase in domain-mean $LWP$ was attributed to the 14 % increase in cloud fraction. Yet, the open cells remain partially un-covered, which prevents a potential transition from the open-cell state to the closed-cell regime.

Mean $A_{cld}$ is increased by 0.15 from 0.18 in the *ctrl* simulation to 0.33 inside the seeded region of the *ship* simulation (Table 2). This translates to a change in total albedo of 0.11 inside the seeded region, which corresponds to a shortwave cloud-radiative effect (SW CRE) of $44\,\mathrm{W\,m}^{-2}$ at an annual mean solar insolation of $404\,\mathrm{W\,m}^{-2}$ at the VOCALS-REx field site. Averaged

over the entire domain the albedo increased by 0.05, which is equivalent to a SW CRE of $20\,\mathrm{W\,m}^{-2}$ exerted over an area of $180 \times 180\,\mathrm{km}^2$.

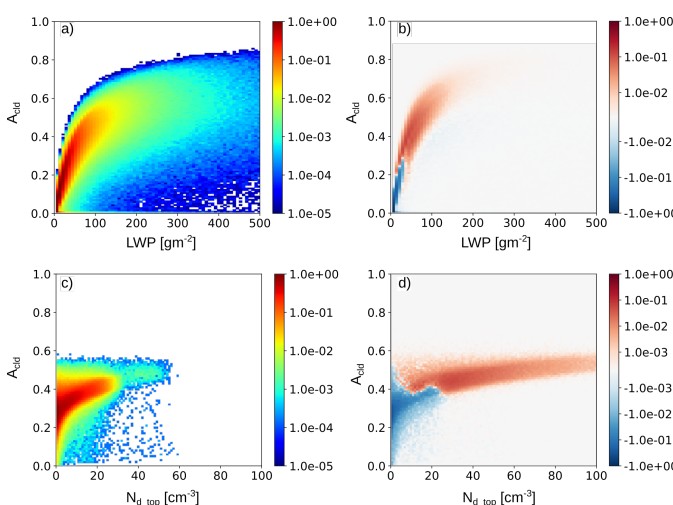

**Figure 4.** Occurrence rate F [%] for the a) liquid water path ($LWP$) versus cloud albedo ($A_{cld}$) phase space and c) the cloud-top droplet number concentration ($N_{d\_top}$) versus $A_{cld}$ phase space. The $N_{d\_top}$-$A_{cld}$ space was sub-filtered for $LWP$ within the range of $40-60\,\mathrm{g\,m}^{-2}$. Results are shown in a) and c) for the last 24 h of the *ctrl* simulation. Absolute changes in F for the *ship* simulation with respect to the *ctrl* simulation are shown in b) and d) respectively. F is normalised to 100 % across the shown phase space. The bin widths for each of which F is defined are $\Delta LWP : 7\,\mathrm{g\,m}^{-2}$, $\Delta N_{d\_top} : 1\,\mathrm{cm}^{-3}$, and $\Delta A_{cld} : 0.01$.



The changes in domain mean albedo were attributed to albedo changes of the detrained cloud sheets spanning the domain between the cell walls. Both, the areal coverage and reflectivity of the detrained cloud sheets increased in the *ship* simulation as compared to the *ctrl* simulation (Table 2). Meanwhile, the cell-wall albedo of 0.6 remained unaffected by the aerosol perturbation. Furthermore, we attributed the simulated change in albedo predominantly to adjustments in macrophysical cloud properties of the detrained cloud regions. Changes in cloud microphysical properties and the associated Twomey (1991) effect were found to be of secondary importance to the change in total albedo.

The increase in cloud fraction alone, while assuming no further changes in in-cloud albedo (i.e. assuming $A_{cld}$ as in the *ctrl* "detrained" and "wall" regions in Table 2), accounted for 91% (100%) of the increase in domain-averaged $A_{cld}$ inside the seeded (unseeded domain). The additional increase in domain-mean $A_{cld}$ within the seeded domain was attributed to the increase in in-cloud albedo of 0.06 within the cloud filaments of the *ship* simulation (Table 2).

Fig. 4 shows the normalised occurrence rate ($F$) within the detrained cloud regions. $F$, and the change in $F$ due to the aerosol perturbation, is shown for each bin within the $LWP$-$A_{cld}$ phase space (Fig. 4a and Fig. 4b respectively). The behaviour of $F$ within the $N_{d\_top}$-$A_{cld}$ space, which was sub-filtered to only include points where in-cloud $LWP$ ranged between $40-60\,\mathrm{g\,m^{-2}}$, is shown in Fig. 4c and Fig. 4d. The behaviour of $F$ for other $LWP$ sub-ranges was found to be qualitatively similar (Fig. S6).

While the increased occurrence of moderate $LWP$ values ($50 \leq LWP < 150\,\mathrm{g\,m^{-2}}$), which was discussed earlier, may coincide with a local increase in $A_{cld}$ (Fig. 4b), the overall decrease in mean in-cloud $LWP$ implied that the increase in $A_{cld}$ could not be caused by $LWP$ adjustments. If anything, in-cloud albedo would be expected to decrease given the reduction in in-cloud $LWP$. Meanwhile, Fig. 4d displayed a clear shift in $F$ towards higher $N_{d\_top}$ associated with locally increased $A_{cld}$. Hence, the increase in in-cloud $A_{cld}$ was attributed to the Twomey (1991) effect within the stratified cloud.

### 3.2.1 Contrasting the cloud response in deep and shallow open cells

Although the areal coverage of the detrained cloud amount between the cell-walls of the open cells increased, which contributed to the brightening of the cloud deck, the highly concentrated aerosol perturbation was insufficient to induce a transition from open to closed cells in these simulations. Aerosols may impact this transition via aerosol-precipitation interactions. Decreases in $N_a$ from $90\,\mathrm{cm^{-3}}$ to $10\,\mathrm{cm^{-3}}$ facilitated a rapid transition from the closed to the open-cell state (Feingold et al., 2015) in previous simulations within the $800\,\mathrm{m}$ deep boundary layer observed during DYCOMS-II. Yet, the reverse transition from the open-cell state to the closed-cell state occurred over far longer time scales, if at all (Wang and Feingold, 2009; Feingold et al., 2015). Nonetheless, strongly concentrated sea salt emissions of $10^{17}$ particles $\mathrm{s^{-1}}$ within the same boundary layer, induced a transition from the open-cell to a filled-in cloud-cell state along the seeding line. Along the seeding line a secondary circulation maintained the cloud layer within the track while depleting the surrounding cloud (Wang et al., 2011).



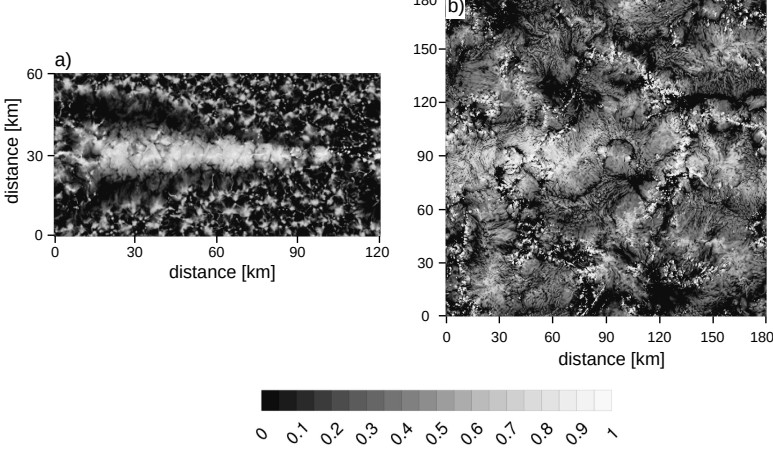

**Figure 5.** a) Cloud albedo field from Wang et al. (2011) Fig. 1a. b) Cloud albedo at 12 UTC on second day of the *ship* simulation. Both simulations were subject to an equal seeding source of $10^{17}$ particles s$^{-1}$.

While a ship track formed in the shallow boundary layer with open-cells between $10-15$ km in size (Fig. 5a), a well-defined track does not form in the deep boundary layer with characteristic cloud cell sizes of $30-40$ km (Fig. 5b). A ship track is also not detected in $N_{d\_top}$ or $R_{reff\_top}$ (Fig.3b and Fig.3c, respectively). The absence of a track in the deep boundary layer is largely attributed to: (i) the large spatial scales of variability within the background cloud state which is determined by the

cloud dynamics and cloud-field organisation, and (ii) the incomplete filling in of the detrained cloud amount between cell-walls, which prevents the transition to a 100% cloud-covered state.

A change in $A_{cld}$ of 0.15, which is of the same magnitude as previously identified in ship tracks (Christensen and Stephens, 2011; Goren and Rosenfeld, 2012; Wang et al., 2011) is found embedded in deep open cells of the *ship* simulation. $N_{d\_top}$ was increased by 167 % and $R_{eff\_top}$ decreased by 14 %. Yet, these effects remain seemingly hidden in the large variability of the

cloud properties governed by the dynamics of the cloud cells. Furthermore, these effects may not easily be attributed to aerosol perturbations via remote sensing techniques of cloud properties as most of the changes in local cloud properties remain within the variability of the system.

Knowing the position and extent of the aerosol perturbation allows one to remove a sufficient amount of variability within the *ship* simulation to obtain a spatially constrained, detectable and attributable response within the cloud properties. As one

averages along the spatial dimension of the aerosol perturbation (coinciding with the x-direction of the simulation domain) the pronounced shift in cloud properties between the seeded and unseeded regions of the *ship* simulation (Fig. 6) is highlighted. However, while changes in total albedo induced within the seeded region may be identified in this manner, the change in total albedo of 0.03 (Table 2) within the detrained cloud regions of the surrounding unseeded domain, would still not be accounted for. Furthermore, changes in $A_{cld}$ within the detrained cloud sheets where found up to 60 km from the emission line, which



has implications for the definition of the truly unperturbed albedo within satellite retrievals of such scenes.

## 4 Implications for aerosol radiative forcing estimates in marine stratocumuli

Estimating the aerosol-induced radiative forcing in low-level marine clouds constitutes a considerable uncertainty in the overall

cloud-radiative forcing of anthropogenic aerosols. Satellite-based changes in CRE estimates due to ship exhaust have remained inconclusive due to the high degree of variability within the natural cloud scene (Peters et al., 2011). GCM estimates provide a wide range of CRE changes between $-0.6\,\mathrm{Wm^{-2}}$ and $-0.07\,\mathrm{Wm^{-2}}$ (Lauer et al., 2007; Righi et al., 2011; Peters et al., 2012; Partanen et al., 2013) due to open-ocean shipping. Furthermore, it remains unclear whether GCMs represent the relevant scales of variability to provide reliable CRE estimates. The analysis of global datasets of ship tracks (Chen et al., 2015) and

volcano plumes (Toll et al., 2017), which have been used as analogues to study the cloud response to anthropogenic emissions, have shown that in the global mean, the cloud response within the tracks largely follows the brightening expected by Twomey (1991). In the global mean, increases and decreases in $LWP$ within the different cloud regimes seem to off-set one another, while many GCMs predict a positive LWP response only (Wang et al., 2012; Ghan et al., 2016; Malavelle et al., 2017; Toll et al., 2017).

In this study we demonstrate that non-negligible amounts of brightening due to anthropogenic shipping emissions may persist in the absence of a clear ship track. In deep open cells, perturbations in $A_{cld}$ were found to be as large as $0.15$ in regions where $\Delta N_{a\_sub}$ is high and as large as $0.08$ when integrated over the whole simulation domain of $180 \times 180\,\mathrm{km^2}$. Furthermore, the induced brightening, which is almost as high as in simulations displaying a pronounced ship track ($\Delta A_{cld} = 0.1$ in Wang et al. (2011)) remains obscured by the variability of the unpolluted cloud, where $LWP$ and $N_{d\_top}$ in itself may differ by an order

of magnitude between convective cell-walls and stratified regions of detrained cloud (Fig. 2).

Furthermore, increases in cloud-scene albedo were attributed to changes in brightness within the stratified, detrained cloud regions covering the boundary layer between convective cell-walls. These detrained cloud regions are optically thin ($\overline{\tau} = 2.8$) and are often referred to as veil clouds. They are connected to the sub-cloud layer aerosol through the convective cell-walls feeding into the detrained cloud regions. In our simulations, these detrained cloud filaments contributed $82\,\%$ to the overall

cloud fraction.

In summary, our results suggest that even though detectable ship tracks are extremely rare in deep boundary layers, an increase in $A_{cld}$ of the order of $0.1$ may persist in deeper boundary layers of open-cell stratocumuli. Furthermore, our simulations suggest that the albedo increase within this regime, which is currently not picked up in ship track analyses, could be driven predominantly by increases in cloud fraction, as opposed to the Twomey (1991) effect.






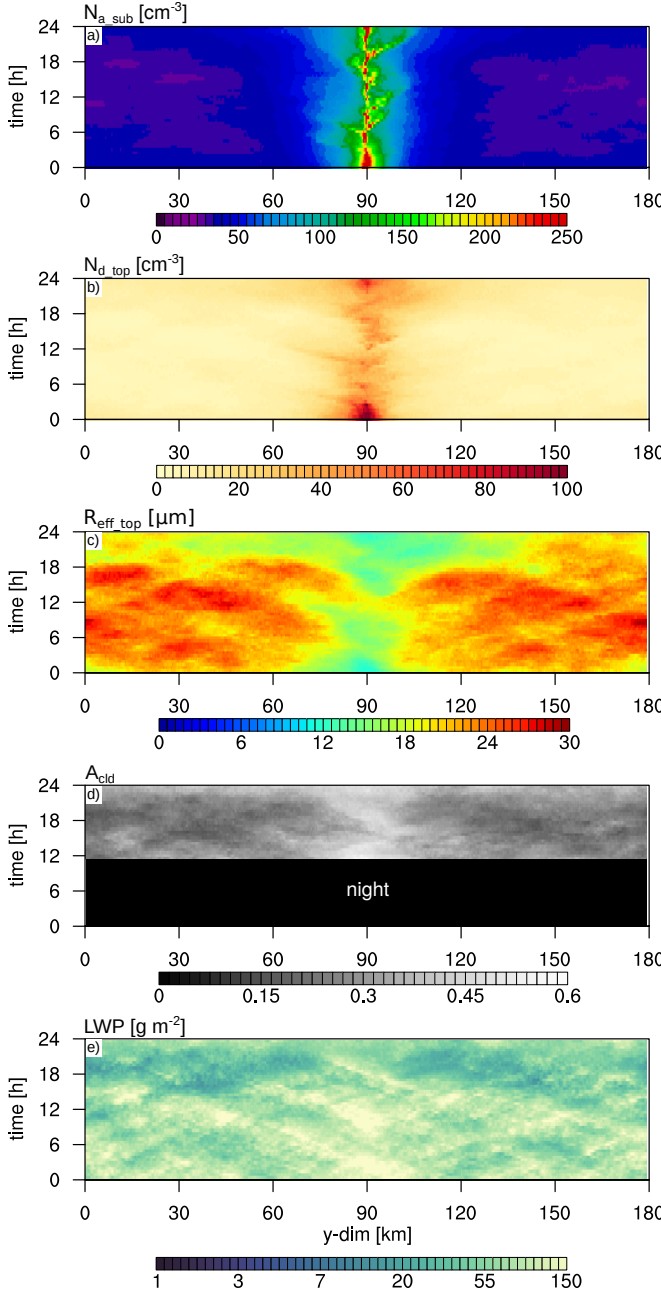

**Figure 6.** Hovmoeller diagrams of a) sub-cloud layer mean aerosol concentration ($N_{a\_sub}$), b) cloud-top cloud droplet number concentration ($N_{d\_top}$), c) cloud-top mean effective cloud droplet radius ($R_{eff\_top}$), d) cloud albedo ($A_{cld}$) and e) liquid water path ($LWP$). Spatial averages were obtained along the emission line dimension (coinciding with x-dimension of simulation domain). Hovmoeller diagrams for the *ctrl* simulation are shown in the supporting material (Fig. S7).





While, these simulations are limited in their generality, they do demonstrate that substantial changes in $A_{cld}$ may occur in optically thin veil clouds and that the aerosol-induced changes in cloud scene albedo may prove extremely difficult to attribute without knowing the spatio-temporal distribution of the aerosol perturbation. Despite significant changes in cloud-scene albedo, an attribution of these changes to an aerosol perturbation using satellite retrievals of cloud-properties and vertically integrated aerosol metrics alone could prove to be extremely difficult in this cloud regime (Fig. 3). Yet, such open-cell cloud scenes with substantial cloud-fraction and a high percentage of veil clouds, occur often (McCoy et al., 2017) and occur in regions of high solar insolation. Therefore, aerosol induced cloud-radiative perturbations within these clouds may be relevant to global estimates of aerosol-cloud-radiative forcing.

Our results strongly motivate further research into the efficacy of aerosol perturbations in deep open-cell stratocumulus. Here we demonstrate, that the aerosol forcing in this regime could be substantial. Yet, for a clear assessment the occurrence rate and magnitude of $A_{cld}$ changes in stratified detrained cloud remnants need to be known. One approach to constrain these aerosol induced perturbations, could be field measurements around known aerosol perturbations. Such measurements would allow the detection and attribution of cloud-radiative effects to aerosol perturbations.

## 5 Conclusions

The analysis of ship tracks and changes in cloud-radiative properties within them, has arguably been an extremely useful framework to develop a mechanistic understanding of aerosol-cloud-radiative interactions, and to constrain the effective cloud-radiative forcing within marine low-level clouds. However, such tracks are extremely rare and tend to form in shallow boundary layers (boundary layer top below $800\,\mathrm{m}$).

At least 70 % of marine stratocumuli form in deeper boundary layers, where distinct ship tracks due to ship emissions are very rarely detected. Here, we assessed in idealised cloud-resolving simulations, whether significant cloud-radiative perturbations persist in a field of deep (boundary layer top at $1.5\,\mathrm{km}$) open-cell stratocumulus, which was observed during RF06 of the VOCALS-REx campaign. Our key-findings are summarised as follows:

1. Albedo changes equivalent to albedo increases in previously observed ship tracks within shallow open-cell stratocumuli were embedded within a stratocumulus deck of deep open cells despite the absence of a spatially coherent structure such as a ship track. The domain-mean total albedo increased by $0.05$ due to a prescribed seeding source (sea salt emission moving at $5\,\mathrm{m\,s^{-1}}$ releasing particles of $300\,\mathrm{nm}$ in size at a rate of $10^{17}$ particles $\mathrm{s^{-1}}$), which translates to a change in th SW CRE of $20\,\mathrm{W\,m^{-2}}$ for an annular solar mean insolation of $404\,\mathrm{W\,m^{-2}}$ at this site.

2. Regional changes in $A_{cld}$ (increase by 0.15), cloud microphysical (167 % increase in $N_{d\_top}$ and 14 % decrease in $R_{eff\_top}$), and macrophysical properties (64% increase in CF and a $5-23$ % decrease in in-cloud $LWP$) within the seeded domain ($\pm 30\,\mathrm{km}$ around the emission line) remain obscured by the naturally occurring variability of cloud field. Reducing the variability of the clouds by averaging along the spatial extend of the aerosol perturbation permitted the detection and attribution of these cloud-radiative effects to the aerosol perturbation. Hence, knowledge of the spatio-



temporal distribution of the aerosol perturbation was found to be necessary for the remote attribution of aerosol effects on cloud-radiative properties within this regime.

3. The simulated cloud brightening was attributed to the brightening of the detrained cloud filaments spanning the regions between the convective cell walls of the open cells. These so-called veil clouds occur frequently in low-level cloud layers and are connected to sub-cloud aerosol sources through the convective cloud cores within the cell walls. Within these clouds the brightening was largely attributed to increases in cloud fraction with a secondary contribution in brightening due to changes in cloud microphysical properties.

*Competing interests.* The authors are not aware of any competing interests.

*Acknowledgements.* We acknowledge the Fund for Innovative Climate and Energy Research grant for the financial support of this research and the high-performance computing support from Yellowstone (ark:/85065/d7wd3xhc). This support was provided by NCAR's Computational and Information Systems Laboratory, sponsored by the National Science Foundation. H. Wang acknowledges support from the U.S. Department of Energy (DOE) Office of Science, Biological and Environmental Research. The Pacific Northwest National Laboratory (PNNL) is operated for DOE by Battelle Memorial Institute under contract DE-AC05-76RLO1830.



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
