# Peer review of "The efficacy of aerosol-cloud-radiative perturbations from near-surface emissions in deep open-cell stratocumulus"

_Atmospheric Chemistry and Physics, 2018_

## Referee Comment (RC1) · Anonymous Referee #1 · 11 Sep 2018

Possner et al. present a modelling study in which the effects of idealised ship emissions on clouds and radiation are examined. Tools and analysis are adequate, and the diligent analysis by Possner et al. allowed for very important insights. A particularly important finding is that a substantial increases in all-sky albedo even relatively far away from the aerosol emission source may be expected, and that there is no simple way of identifying these changes given the large natural variability of the cloud properties. With a known location of the emission source, in turn, detection and attribution seems feasible.

The study is very relevant to the readership of Atmos. Chem. Phys. and should be

published after some minor revisions.

The most important one is that the authors need to clarify better than they did which quantities are reported as all-sky, and which as cloudy-sky averages. Some further mistakes or unclear aspects that I list below also should be corrected.

Specific comments

p1l19 – the forcing is for the anthropogenic perturbation of the aerosol

p2l5 – invert the order of the Durkee references

p2l15 – Toll rather did conclude that the LWP change was small

p2l19 – correct "Agency" reference

p2l31 – unit missing (cm-3?)

p2l32 – reference to Platnick and Twomey is missing

Table 1 – interquartile range: of the temporal variability of the domain-mean quantities? should be specified. Are LWP and Nd in-cloud or all-sky in both, observations and model results?

p3l14 – The quantities in Table 1 are not numerous, and don't include boundary layer properties

p4l20 – a reference for the Quickscat data source would be good

p5l10 – for completeness, the symbols $\tau$, h, and z could be explained, too. $\rho_w$ presumably is a constant?

p6l6 – how do the authors come to the conclusion that these are "realistic"?

p7l6 – if the authors want to make this point, they should consider listing the observed range in Table 1.

Fig. 2 – how is cloud top defined?

Table 2 – the caption should report the definition of "ship-seeded" / "ship-unseeded" as well as "detrained" and "wall"

p8 l14 – the plume with Na> 1000 cm-3 seems to me much narrower than 60 km. I would guess, only a few km

p9l4 – were → was

p10 last line – reduction compared to what? to the ctrl simulation?

p12l7-9 – I have difficulties following this argument. What is meant by "domain-average Acld"? Is it what is labeled "albedo" in Table 2? It is hard to understand that the increase in in-cloud albedo by 5

p12l17 – this decrease in in-cloud LWP is not documented in the Tables or figures. Or is the area covered by detrained + wall the entire cloudy area?

p12l20 – given the relatively small mean increases in Nd, and the fact that albedo is much less sensitive to Nd changes than to L changes, is this plausible? Or is the distribution of the relative changes of both quantities relevant?

p14l5 – it probably rather is "satellite-based estimates in CRE changes"

p14l16 – maybe specify "even in the absence"?

p16l1 – superfluous comma

p16l26 – the

p16l27 - "annual mean insolation"

p16l31 – extent

Supplement

Fig. 2a – this seems to me an awkward definition of a CDF. I am used to CDFs that increase monotonically and asymptotically approach 1 (as in Fig. 5, in fact).

Fig. 3a – it would be good to indicate the ship position here, as well as the past ship track

Fig. 6 – this figure seems to be wrong. I believe the authors intended to show Nd_top vs. AcId histograms in all rows.

[Figure]

---

## Referee Comment (RC2) · Anonymous Referee #2 · 18 Sep 2018

General comments: Possner et al present an interesting modeling study in which the effect of ship emissions on cloud microphysical and macrophysical properties of deep open cells is examined. Based on field campaign measurements, and previous modeling study of Wang et al 2010, Possner et al. show that despite the lack of typical linear ship tracks, the cloud adjustments can be significantly larger than one would expect. The manuscript is well written and the analysis support the authors conclusions. I recommend the manuscript to be published after minor revisions.

Specific comments: 1. The authors cite that 70% of marine Sc form in deep boundary layers (p16l19; citation needed, e.g., Muhlbauer et al., 2014). Do the 70% compose

mainly open cells? How much of the 70% are closed cells? This needs to be mentioned in order to asses the global effect of ship emissions in deep open cells.

2. The domain mean increase in albedo is function of the domain size. For a smaller domain, the increase would be larger, and for larger domain size, smaller. Therefore, domain mean increase in albedo of 0.05 is somewhat arbitrary. If the authors can estimate the density of ship tracks in a given regions with frequently observed deep open cells, a more meaningful value of regional mean increase in albedo can be estimated.

3. The authors claim in the abstract that changes in cloud-radiative properties are masked by the natural variability. What is the meaning of natural variability in this context? The abstract further says that the above can be overcome by utilizing the spatio-temporal distribution of the aerosol perturbation. However, in Figure 3 the aerosol plume can be easily seen in Nd, which serves as a tracer to where one can expect cloud adjustment. This can be used in observational studies.

4. The authors should improve the description of the tables: Table 1: The caption says the simulated values are domain mean. These values are compared with RF06, which seems to be in-clouds values (for LWP at least). Clarification is needed. Table 2: The left column is unclear. What is the difference between ship, ship-seeded and ship-unseeded? (ship-unseeded is not mentioned anywhere else in the text). Under the CF column, how CF can be not 100% inside walls? given that walls are defined by ascending air? Is the wall CF is the fraction of walls out of the total CF/domain? If so it means that there is also a dynamical adjustment.

5. I recommend to elaborate more in the introduction on previous studies that attempted to quantify the regional effect of ship tracks (e.g., Schrier et al. 2006, 2007, Peters et al. 2011).

6. The simulation assumes an idealized case with no perpendicular winds. I assume that most ship tracks don't have head/tail winds, rather side winds. Would the wind direction relative to the emission source increase/decrease the regional area that is

affected by the emissions? This should discussed.

Technical corrections: Section 2.2: What was the duration of the simulations?

P8l6: Remove "and".

P8l15: Any statistical tests were done to determine the 30km band around the emission line? It is mentioned that inside this region Na_sub are elevated, but by how much? I also would expect the plume to expand and dilute as it gets more mature, and not being fixed.

P12l7-10: This paragraph is not clear.

P12l28: Observational studies showed ship tracks closing open cells (e.g., Goren and Rosenfled 2012 where at least part of the open cells seems to be deep, based on the cells spatial scale; Christensen and Stephens 2012). While simulations does not show a reverse transition, observational evidences should be provided as well.

P16l17: Consider changing "such tracks" to "linear shaped tracks", and to add that they are rare in deep boundary layers in comparison to shallow boundary layers (reference is needed).

P16l28-29: How do the fractional percentage calculated? From which table?

P16l26 the → the.

P16l27 annular → annual.

Figure 5: In the caption the boundary layer depth for each of the simulations should be provided (i.e., shallower in Wang et al 2011).

Figure 6: In order to cover also the night time in Figure 6d, consider replacing (or adding) cloud optical thickness with cloud albedo?

Supplementary: Caption 3: What is "ship_open"?

Caption 6: remove "and" in line 3. Caption is not clear. X axis label is not consistent.

---

## Author Comment (AC1) · 22 Sep 2018

Possner et al. present a modelling study in which the effects of idealised ship emissions on clouds and radiation are examined. Tools and analysis are adequate, and the diligent analysis by Possner et al. allowed for very important insights. A particularly important finding is that a substantial increases in all-sky albedo even relatively far away from the aerosol emission source may be expected, and that there is no simple way of identifying these changes given the large natural variability of the cloud properties. With a known location of the emission source, in turn, detection and attribution seems feasible.

The study is very relevant to the readership of Atmos. Chem. Phys. and should be published after some minor revisions.
We thank the reviewer for the assessment and have addressed all issues raised in the revised manuscript. Please find our response to each of the individual comments below.

The most important one is that the authors need to clarify better than they did which quantities are reported as all-sky, and which as cloudy-sky averages. Some further mistakes or unclear aspects that I list below also should be corrected.
Domain means have been clarified with respect to when they were taken over all sky, cloudy sky or clear sky. To this end the captions of Table 1 and Table 2 were revised and minor edits were undertaken in the main text of the manuscript.

Specific comments

p1l19 – the forcing is for the anthropogenic perturbation of the aerosol
This was corrected in manuscript.
p2l5 – invert the order of the Durkee references
Done.
p2l15 – Toll rather did conclude that the LWP change was small
The increase in LWP was quantified in main text for accuracy. According to Fig.2 of Toll et al. (2017), the increase in the precipitating regime was between $16 - 24\%$.
p2l19 – correct "Agency" reference
Done.
p2l31 – unit missing (cm-3?)
Added.
p2l32 – reference to Platnick and Twomey is missing
Added.
Table 1 – interquartile range: of the temporal variability of the domain-mean quantities? should be specified. Are LWP and Nd in-cloud or all-sky in both, observations and model results?
The caption has been revised for clarity. Domain-mean and in-cloud Nd are given in the table (correspondingly for observations and simulations). The interquartile range covers the spatial and temporal variability of the cloud field. This is now clarified in the revised caption.
p3l14 – The quantities in Table 1 are not numerous, and don't include boundary layer properties
The sentence has been rephrased.
p4l20 – a reference for the Quickscat data source would be good
Added.

p5l10 – for completeness, the symbols $\tau$ , h, and z could be explained, too. $\rho$ w presumably is a constant?

A definition of all variables has been added.

p6l6 – how do the authors come to the conclusion that these are "realistic"?

The cell sizes of open cells observed during VOCALS-REx were around 30-40km (p3L15), which is consistent with Fig.1d. Rephrased "realistic" with "observed".

p7l6 – if the authors want to make this point, they should consider listing the observed range in Table 1.

A reference to Fig. S2, where the range of observed Rcb is shown, has been added.

Fig. 2 – how is cloud top defined?

Reference to definition of cloud top given in revised caption of Table 1 has been added.

Table 2 – the caption should report the definition of "ship-seeded" / "ship-unseeded" as well as "detrained" and "wall"

The definitions have been added to the caption. Entries are rephrased as "seeded" and "unseeded" also.

p8 l14 – the plume with Na> 1000 cm-3 seems to me much narrower than 60 km. I would guess, only a few km

The 60 km refers to the length and not the width of the plume. This has now been clarified in the revised manuscript.

p9l4 – were → was

Done.

p10 last line – reduction compared to what? to the ctrl simulation?

The decrease in Nd_top is with respect to the cell-wall cores. This has been clarified in the revised manuscript.

p12l7-9 – I have difficulties following this argument. What is meant by "domain-average Acld"? Is it what is labeled "albedo" in Table 2? It is hard to understand that the increase in in-cloud albedo by 5

It was meant to be "all-sky" albedo. The revised caption of Table 2 together with the renaming of "albedo" as $A_{all}$ and $A_{cld}$ should clarify this discussion in the revised manuscript.
The calculation is that:
CF_filament_ship*Acld_filament_ctrl+CF_wall_ship*Acld_wall_ctrl=0.9*
CF_filament_ship*Acld_filament_ship+CF_wall_ship*Acld_wall_ship
where e.g. "CF_filament_ship" corresponds to the cloud fraction entry of Table 2 for the cloud filaments of the ship simulation.

p12l17 – this decrease in in-cloud LWP is not documented in the Tables or figures. Or is the area covered by detrained + wall the entire cloudy area?

Yes, the table entries under filament and wall are cloud-mean only, and one can see the decrease from detrained ctrl to detrained seeded of the ship simulation. This is now clarified in the revised caption of Table 2.

p12l20 – given the relatively small mean increases in Nd, and the fact that albedo is much less sensitive to Nd changes than to L changes, is this plausible? Or is the distribution of the relative changes of both quantities relevant?

The increase in in-cloud $A_{cld}$ of the filaments was attributable to changes in Nd and, therefore, the Twomey effect. In-cloud mean LWP within the filaments is found to decrease and, therefore, cannot contribute to the brightening. However, it should be noted that 90% of the increase in domain-mean $A_{cld}$ does not come from brightened clouds, but is attributed to the increase in cloud fraction assuming an equally bright cloud distribution. This should now be clearer in the revised manuscript given the more careful definition of all-sky and in-cloud mean quantities.

p14l5 – it probably rather is "satellite-based estimates in CRE changes"

Done.

p14l16 – maybe specify "even in the absence"?

We decided to keep the sentence as is.

p16l1 – superfluous comma

Removed.
p16l26 – the
Done.
p16l27 - "annual mean insolation"
Changed.
p16l31 – extent
Changed.

Supplement
Fig. 2a – this seems to me an awkward definition of a CDF. I am used to CDFs that increase monotonically and asymptotically approach 1 (as in Fig. 5, in fact).
This was a misnomer of the Figure. Indeed, the probability density function (PDF) is shown. This has been corrected.

Fig. 3a – it would be good to indicate the ship position here, as well as the past ship track
We have played around with super-imposing the aerosol perturbation in a transparent layer over the cloud fields and decided not to do it for clarity of the figures. The information is instead displayed separately in Fig 3 of the main manuscript (Fig 3a - aerosol plume and Fig 3b - Nd_top field) separately.
Fig. 6 – this figure seems to be wrong. I believe the authors intended to show Nd_top vs. Acld histograms in all rows.
Thank you for pointing this out. It has been corrected in the revised supplement.

---

## Author Comment (AC2) · 22 Sep 2018

General comments: Possner et al present an interesting modeling study in which the effect of ship emissions on cloud microphysical and macrophysical properties of deep open cells is examined. Based on field campaign measurements, and previous modeling study of Wang et al 2010, Possner et al. show that despite the lack of typical linear ship tracks, the cloud adjustments can be significantly larger than one would expect. The manuscript is well written and the analysis support the authors conclusions. I recommend the manuscript to be published after minor revisions.

We thank the reviewer for the evaluation and have addressed all comments in the revised manuscript. Individual responses to each of the issues raised are inserted below.

Specific comments: 1. The authors cite that 70% of marine Sc form in deep boundary layers (p16l19; citation needed, e.g., Muhlbauer et al., 2014). Do the 70% compose mainly open cells? How much of the 70% are closed cells? This needs to be mentioned in order to asses the global effect of ship emissions in deep open cells.

From Table 4 of Muhlbauer et al. (2014) we know that open-cell and disorganised stratocumulus clouds occur far more often than the closed-cell regime. A further height-dependent split of the frequency of occurrence in open, closed and disorganised stratocumuli was not obtained by Muhlbauer et al. (2014) and is not available to us. While the authors agree that this would be needed for a global assessment of the radiative impact of ship emissions, it is beyond the scope of this study, which focuses on the feedback mechanisms in such regimes and explores the potential for radiative impacts that were previously unexplored and unquantified.

The issue of relative regime occurrence was previously only mentioned in the introduction of the submitted manuscript (P2L26ff):
"Yet, over 70% of stratocumulus clouds are found in deeper boundary layers (Muhlbauer et al., 2014). The potential for albedo changes is particularly high in the open-cell, and disorganised stratocumulus regimes, which occur more frequently in the sub-tropics than in the closed cells regime (Muhlbauer et al., 2014)." A more quantitative statement has now also been added to the conclusions (including the reference to Muhlbauer et al., 2014).

2. The domain mean increase in albedo is function of the domain size. For a smaller domain, the increase would be larger, and for larger domain size, smaller. Therefore, domain mean increase in albedo of 0.05 is somewhat arbitrary. If the authors can estimate the density of ship tracks in a given regions with frequently observed deep open cells, a more meaningful value of regional mean increase in albedo can be estimated.

It should be noted that the initial design of the emission flux in Wang et al (2011) was taken in relation to the suggested flux of sea salt for potential marine brightening applications. The sea salt emission flux chosen is indeed proportional to the domain size. The realization or injection strategy does matter, as discussed in by Wang et al. (2011). To address the relevance of the emission strategy with respect to ship traffic the following paragraph was added to the manuscript at P14L20:
Furthermore, while these simulations are highly idealised in their setup, they do not necessarily reflect unrealistic emission conditions. The prescribed ship is assumed to travel periodically along an identical emission line without any crosswind, which may alter the plume size or dilute

emissions more effectively.  Within the 48 h simulation, a total of 5 ships traverse the 180x180 km$^2$ domain repeatedly at a constant sailing speed of 5ms$^{-1}$, and the cloud response to their combined emissions is assessed. Throughout most of the North Pacific a shipping density of around 30 ships per 100 km$^2$ per year is observed (MarineTraffic, 2018). Assuming a speed of 5ms$^{-1}$ (or even 10ms$^{-1}$), such a density corresponds to an estimated number of 116 (58) ships within the simulation domain on average. Within the North Atlantic, the higher density of ships could even correspond to over 400 (200) ships within a 180x180 km$^2$ domain (MarineTraffic, 2018).  Therefore, our emission scenario is equivalent to merely $1 - 9\%$ of these ships contributing to increased CCN concentrations within the seeded domain.

Furthermore, to clearly highlight that the brightening occurred throughout the domain (though most brightening remained constrained to the seeded domain). The following figure (Fig. S6) was added to the appendix and referenced on P11L16:

[Figure]

**Figure 6.** Across-track difference in all-sky albedo ($A_{all}$) between the *ship* and *ctrl* simulation averaged over the last 24 h of both simulations. Solid grey line denotes the location of the emission line of the ship, while grey dashed lines mark the seeded domain ($\pm$30 km from emission line).

3. The authors claim in the abstract that changes in cloud-radiative properties are masked by the natural variability. What is the meaning of natural variability in this context?
For clarification we rephrased "natural variability" with "naturally occurring variability" (i.e., the variability occurring within the cloud field without an anthropogenic aerosol perturbation).

The abstract further says that the above can be overcome by utilizing the spatio-temporal distribution of the aerosol perturbation. However, in Figure 3 the aerosol plume can be easily seen in Nd, which serves as a tracer to where one can expect cloud adjustment. This can be used in observational studies.
In Figure 3 only small parts of the entire seeded domain are characterised by perturbations in cloud-top Nd of 100 cm$^{-3}$ or above. Throughout most of the seeded domain, where CCN concentrations are increased by a factor 2, or even in regions where the CCN concentrations exceed 100 cm$^{-3}$, Nd is well within the background variability. Furthermore, the higher concentrations of Nd~100cm$^{-3}$ are neither unrealistically high for a naturally occurring background, nor spatially coherent to be picked up as an unambiguous marker of an anthropogenic perturbation. Finally, if one were to use only sub-regions where Nd is increased, one would likely miss the full radiative response simulated here

for open cells, where the predominant forcing is due to an increase in cloud cover of the cloud filaments with a low Nd. For these reasons the authors decide to keep the statement that our simulations indicate that the spatio-temporal distribution of the aerosol is needed to determine the full extent of the cloud-radiative impact by the ship emissions in this regime.

4. The authors should improve the description of the tables: Table 1: The caption says the simulated values are domain mean. These values are compared with RF06, which seems to be in-clouds values (for LWP at least). Clarification is needed. Table2: The left column is unclear. What is the difference between ship, ship-seeded and ship-unseeded? (ship-unseeded is not mentionedanywhere else in the text). Under the CF column, how CF can be not 100% inside walls? given that walls are defined by ascending air? Is the wall CF is the fraction of walls out of the total CF/domain? If so it means that there is also a dynamical adjustment.

We agree with the reviewer that the captions provided insufficient information. Both captions have been revised and the LHS column of Table 2 is now renamed in the revised manuscript.

5. I recommend to elaborate more in the introduction on previous studies that at-
tempted to quantify the regional effect of ship tracks (e.g., Schrier et al. 2006, 2007,
Peters et al. 2011).
Missing references have been added to the manuscript.

6. The simulation assumes an idealized case with no perpendicular winds. I assume that most ship tracks don't have head/tail winds, rather side winds. Would the wind direction relative to the emission source increase/decrease the regional area that is affected by the emissions? This should discussed.
Yes, the model domain was purposely aligned with the wind direction in order to identify a potential linear structure of ship tracks and compare with the previous study (Wang et al., 2011) for a shallow boundary layer. This is incorporated into the new paragraph included in the revised manuscript, which is presented under comment #2.

Technical corrections: Section 2.2: What was the duration of the simulations?
P4L20: Both simulations were run for 48 hours.
P8l6: Remove "and".
"A comma was introduced to clarify the sentence."
P8l15: Any statistical tests were done to determine the 30km band around the emission line? It is mentioned that inside this region Na_sub are elevated, but by how much? I also would expect the plume to expand and dilute as it gets more mature, and not being fixed.
The seeded domain was conservatively identified as the corridor, where CCN concentrations were increased and the meandering plume, consisting of the super-position of 5 consecutive ships, remained within the bounds of this region. We agree that for an individual ship one would expect a widening of the plume with distance. In previous analyses performed for this study we have quantified the core plume in these simulations where Na was outside 3 standard deviations of the background concentration, but this constrained the analysis to a rather narrow region of highest concentration around the emission line. No further insight was obtained from these results and they were therefore omitted in the manuscript. We therefore felt that it was more insightful to distinguish in the analysis between a region where CCN concentrations were elevated and a region where no increase in CCN was detected.

P12l7-10: This paragraph is not clear.
The paragraph has been rephrased for clarity.
P12l28: Observational studies showed ship tracks closing open cells (e.g., Goren and Rosenfled 2012 where at least part of the open cells seems to be deep, based on the cells spatial scale;

Christensen and Stephens 2012). While simulations does not show a reverse transition, observational evidences should be provided as well.

The "large open cells" in Goren & Rosenfeld (2012) are estimated to be around 20-25 km, which is roughly half the size of the cells simulated here. Furthermore, observational evidence (Durkee et al 2000b, Toll et al 2017, Chen et al 2015, Christensen & Stephens 2012) suggests that ship tracks in boundary layers deeper than 1 km are extremely unlikely. Therefore, it is not surprising to see no track-like structure in these simulations. However, it seems that a similar process occurs in deep boundary layers, where large open cells are partially filled in, but the filaments never stretch across the entire cell, which would then allow it to recover (given sufficient mixing generated through cloud-top cooling to overcome sub-cloud stability). The Goren and Rosenfeld (2012) reference was added to the manuscript (P12L33).

P16l17: Consider changing "such tracks" to "linear shaped tracks", and to add that they are rare in deep boundary layers in comparison to shallow boundary layers (reference is needed).

Rephrasing has been done and references from the introduction are repeated here.

P16l28-29: How do the fractional percentage calculated? From which table?

The paragraph was rephrased slightly such that now all numerical results are merely summarised here without reference, but are referenced to corresponding figures and tables in section 3.

P16l26 the → the.

Done.

P16l27 annular → annual.

Done.

Figure 5: In the caption the boundary layer depth for each of the simulations should be provided (i.e., shallower in Wang et al 2011).

The information was added.

Figure 6: In order to cover also the night time in Figure 6d, consider replacing (or adding) cloud optical thickness with cloud albedo?

As only day-time values in $A_{all}$ and $A_{cld}$ are considered throughout the paper, the night-time values are not shown here for consistency. Although they are diagnosed, we omitted them in the averaging process, as they have no physical meaning but affect the temporal mean due to the diurnal cycle (clouds are optically thicker at night).

Supplementary: Caption 3: What is "ship_open"?

This typo has been removed.

Caption 6: remove "and" in line 3. Caption is not clear. X axis label is not consistent.

This has been changed.

---

## Author Response (AR1)

Possner et al. present a modelling study in which the effects of idealised ship emissions on clouds and radiation are examined. Tools and analysis are adequate, and the diligent analysis by Possner et al. allowed for very important insights. A particularly important finding is that a substantial increases in all-sky albedo even relatively far away from the aerosol emission source may be expected, and that there is no simple way of identifying these changes given the large natural variability of the cloud properties. With a known location of the emission source, in turn, detection and attribution seems feasible.

The study is very relevant to the readership of Atmos. Chem. Phys. and should be published after some minor revisions.
We thank the reviewer for their assessment and have addressed all issues raised in the revised manuscript. Please find a repy to each individual comment below.

The most important one is that the authors need to clarify better than they did which quantities are reported as all-sky, and which as cloudy-sky averages. Some further mistakes or unclear aspects that I list below also should be corrected.
Domain means have been clarified with respect to when they were taken over all-sky, cloudy sky or clear sky. To this end the captions of Table 1 and Table 2 were revised and minor edits were undertaken in the main text of the manuscript.

Specific comments

p1l19 – the forcing is for the anthropogenic perturbation of the aerosol
This was corrected in manuscript.
p2l5 – invert the order of the Durkee references
Done.
p2l15 – Toll rather did conclude that the LWP change was small
The increase in LWP was quantified in main text for accuracy. According to Fig.2 of Toll et al 2017 increase in the precipitating regime was between 16 – 24%.
p2l19 – correct "Agency" reference
Done.
p2l31 – unit missing (cm-3?)
Added.
p2l32 – reference to Platnick and Twomey is missing
Added.
Table 1 – interquartile range: of the temporal variability of the domain-mean quantities? should be specified. Are LWP and Nd in-cloud or all-sky in both, observations and model results?
The caption has been revised for clarity. Domain-mean and in-cloud Nd are given in the table (correspondingly for observations and simulations). The interquartile range covers the spatial and temporal variability of the cloud field. This is now clarified in the revised caption.
p3l14 – The quantities in Table 1 are not numerous, and don't include boundary layer properties
Sentence has been rephrased.
p4l20 – a reference for the Quickscat data source would be good
Added.

p5l10 – for completeness, the symbols $\tau$ , h, and z could be explained, too. $\rho$ w presumably is a constant?

A definition of all variables was added.

p6l6 – how do the authors come to the conclusion that these are "realistic"?

The cell sizes of open cells observed furing VOCALS Rex were around 30-40km (p3L15), which is consistent with Fig.1d. Rephrased "realistic" with "observed".

p7l6 – if the authors want to make this point, they should consider listing the observed range in Table 1.

A reference to Fig.S2, where the range of observed Rcb is shown, was added.

Fig. 2 – how is cloud top defined?

Reference to definition of cloud top given in revised caption of Table 1 was added.

Table 2 – the caption should report the definition of "ship-seeded" / "ship-unseeded" as well as "detrained" and "wall"

The definitions were added to the caption. Entries were rephrased as "seeded" and "unseeded" also.

p8 l14 – the plume with Na> 1000 cm-3 seems to me much narrower than 60 km. I would guess, only a few km

The 60km refer to the length and not the width of the plume. This was clarified in the revised manuscript.

p9l4 – were $\rightarrow$ was

Done.

p10 last line – reduction compared to what? to the ctrl simulation?

The decrease in Nd_top is with respect to the cell wall cores. This has been clarified in the revised manuscript.

p12l7-9 – I have difficulties following this argument. What is meant by "domain-average Acld"? Is it what is labeled "albedo" in Table 2? It is hard to understand that the increase in in-cloud albedo by 5

The revised caption in Table 2 together with the renaming of "albedo" as Aall and Acld as appropriate should clarify this discussion in the revised manuscript.

The calculation is that:

$$CF\_filament\_ship*Acld\_filament\_ctrl+CF\_well\_ship*Acld\_wall\_ctrl=0.9*$$
$$CF\_filament\_ship*Acld\_filament\_ship+CF\_well\_ship*Acld\_wall\_ship$$

where e.g. "CF_filament_ship" corresponds to the cloud fraction entry of Table 2 for the cloud filaments of the ship simulation.

p12l17 – this decrease in in-cloud LWP is not documented in the Tables or figures. Or is the area covered by detrained + wall the entire cloudy area?

Yes, the table entries under filament and wall are cloud-mean only and one can see the decrease from detrained ctrl to detrain seeded of the ship simulation. This is now clarified in the revised caption of table 2.

p12l20 – given the relatively small mean increases in Nd, and the fact that albedo is much less sensitive to Nd changes than to L changes, is this plausible? Or is the distribution of the relative changes of both quantities relevant?

The increase in in-cloud Acld of the filaments was attributable to changes in Nd and the Twomey effect. In-cloud mean LWP within the filaments is found to decrease and can therefore not contribute to the brightening. However, it should be noted that 90% of the increase in domain-mean Acld does not come from brighter clouds, but is attributed to the increase in cloud-fraction assuming an equally bright cloud distribution. This should now be clearer in the revised manuscript given the more carefull defintion of all-sky and in-cloud means.

p14l5 – it probably rather is "satellite-based estimates in CRE changes"

Done.

p14l16 – maybe specify "even in the absence"?

We decided to keep the sentence as is.

p16l1 – superfluous comma

Removed.

p16l26 – the
Done.
p16l27 - "annual mean insolation"
Changed.
p16l31 – extent
Changed.

Supplement
Fig. 2a – this seems to me an awkward definition of a CDF. I am used to CDFs that increase monotonically and asymptotically approach 1 (as in Fig. 5, in fact).
This was a mis-noma of the Figure. Indeed the probability density function (PDF) is shown. This has been corrected.

Fig. 3a – it would be good to indicate the ship position here, as well as the past ship track
We have played around with super-imposing the aerosol perturbation in a transparent layer over the cloud fields and decided for clarity of the figures not to do it. The information is instead displayed separately in Fig 3 of the main manuscript (Fig 3a - aerosol plume and Fig 3b - Nd_top field) separately.
Fig. 6 – this figure seems to be wrong. I believe the authors intended to show Nd_top vs. Acld histograms in all rows.
Thank you for pointing this out. The Figure has been corrected in the revised supplement.

**Authors response to: "Interactive comment on "The efficacy of aerosol-cloud-radiative perturbations from near-surface emissions in deep open-cell stratocumulus" by Anna Possner et al.**

Anonymous Referee #2

General comments: Possner et al present an interesting modeling study in which the effect of ship emissions on cloud microphysical and macrophysical properties of deep open cells is examined. Based on field campaign measurements, and previous modeling study of Wang et al 2010, Possner et al. show that despite the lack of typical linear ship tracks, the cloud adjustments can be significantly larger than one would expect. The manuscript is well written and the analysis support the authors conclusions. I recommend the manuscript to be published after minor revisions.

We thank the reviewer for their evaluation and have addressed their comments in the revised manuscript. Individual responses to each of the issues raised are inserted below.

Specific comments: 1. The authors cite that 70% of marine Sc form in deep boundary layers (p16l19; citation needed, e.g., Muhlbauer et al., 2014). Do the 70% compose mainly open cells? How much of the 70% are closed cells? This needs to be mentioned in order to asses the global effect of ship emissions in deep open cells.

From Table 4 of Muhlbauer et al 2014 we know that open-cell and disorganised stratocumulus clouds occur far more often than the closed-cell regime. A further height-dependend split of the frequency of occurrence in open, closed and disorganised stratocumuli was not obtained by Muhlbauer et al 2014 and is not available to the authors. While the authors agree that this would be needed for a global assessment of the radiative impact of ship emissions, it is beyond the scope of this study which focused on the feedback mechanisms in such regimes and explored the potential for radiative impacts, which remained previously unexplored and unquantified.

The issue of relative regime occurrence was previously only mentioned in the introduction of the submitted manuscript (P2L26ff):
"Yet, over 70% of stratocumulus clouds are found in deeper boundary layers (Muhlbauer et al., 2014). The potential for albedo changes is particularly high in the open-cell, and disorganised stratocumulus regimes, which occur more frequently in the sub-tropics, than in the closed cells regime (Muhlbauer et al., 2014)." A more quantitative statement has now also been added to the conclusions (including reference to Muhlbauer et al 2014).

2. The domain mean increase in albedo is function of the domain size. For a smaller domain, the increase would be larger, and for larger domain size, smaller. Therefore, domain mean increase in albedo of 0.05 is somewhat arbitrary. If the authors can estimate the density of ship tracks in a given regions with frequently observed deep open cells, a more meaningful value of regional mean increase in albedo can be estimated.

The following paragraph on this, which incorporates the concerns raised under point 6 has been added to the manuscript at P14L20:
"Furthermore, while these simulations are highly idealised in their setup, they do not necessarily reflect unrealistic emission conditions. The prescribed ship is assumed to travel periodically along an identical emission line without any cross-wind, which may reduce the plume size or dilute emissions more effectively. Within the 48h simulation, a total of 5 ships traverse the $180 \times 180$ km2 domain and the cloud response to their combined emissions is assessed. Throughout most of the North Pacific a shipping density of around 30 ships per 100 km2 per year is observed

(MarineTraffic). Assuming a speed of 10 m s−1 , such a density corresponds to an estimated number of 58 ships within the simulation domain on average. Within the North Atlantic, the higher density of ships could even correspond to over 200 ships within a 180 × 180 km 2 domain (MarineTraffic). Therefore, our emission scenario is equivalent to merely ⬜ 10 % (3 %) of these ships contributing to increased CCN concentrations within the seeded domain."

3. The authors claim in the abstract that changes in cloud-radiative properties are masked by the natural variability. What is the meaning of natural variability in this context?
For clarification we rephrased "natural variability" with "naturally occurring variability" (i.e. the variability occurring within the cloud field without an anthropogenic aerosol perturbation).

The abstract further says that the above can be overcome by utilizing the spatio-temporal distribution of the aerosol perturbation. However, in Figure 3 the aerosol plume can be easily seen in Nd, which serves as a tracer to where one can expect cloud adjustment. This can be used in observational studies.
In Figure 3 only small parts of the entire seeded domain are characterised by perturbations in cloud-top Nd of 100 cm-3 or above. Throughout most of the seeded domain, where CCN concentrations are increased by a factor 2, or even in regions were the CCN concentrations exceed 100cm-3, Nd is well within the background variability. Furthermore, the higher concentrations of Nd~100cm-3 are neither unrealistically high for a naturally occuring background, nor spatially coherent to be picked up as an unambiguous marker of an anthropogenic perturbation. Finally, if one were to use only sub-regions where Nd is increased, one would likely miss the full radiative response simulated here for open-cells, where the predominant forcing seems due to an increase in cloud cover of the cloud filaments where Nd is low. For these reasons the authors remain with the statement that our simulations indicate that the spatio-temporal distribution of the aerosol is needed to determine the full extent of the cloud-radiative impact by the ship emissions in this regime.

4. The authors should improve the description of the tables: Table 1: The caption says the simulated values are domain mean. These values are compared with RF06, which seems to be in-clouds values (for LWP at least). Clarification is needed. Table2: The left column is unclear. What is the difference between ship, ship-seeded and ship-unseeded? (ship-unseeded is not mentionedanywhere else in the text). Under the CF column, how CF can be not 100% inside walls? given that walls are defined by ascending air? Is the wall CF is the fraction of walls out of the total CF/domain? If so it means that there is also a dynamical adjustment.

We agree with the reviewers, that the captions provided insufficient information. Both captions have been revised and the LHS column of Table 2 renamed in the revised manuscript.

5. I recommend to elaborate more in the introduction on previous studies that attempted to quantify the regional effect of ship tracks (e.g., Schrier et al. 2006, 2007, Peters et al. 2011).
Missing references were added to the manuscript.

6. The simulation assumes an idealized case with no perpendicular winds. I assume that most ship tracks don't have head/tail winds, rather side winds. Would the wind direction relative to the emission source increase/decrease the regional area that is affected by the emissions? This should discussed.
This is incorporated into the new paragraph included in the revised manuscript, which is presented under point 2.

Technical corrections: Section 2.2: What was the duration of the simulations?

P4L20: Both simulations were run for 48h.

P8l6: Remove "and".

"A comma was introduced to clarify the sentence."

P8l15: Any statistical tests were done to determine the 30km band around the emission line? It is mentioned that inside this region Na_sub are elevated, but by how much? I also would expect the plume to expand and dilute as it gets more mature, and not being fixed.

The seeded domain was conservatively identified as the corridor, where CCN concentrations were increased and the meandering plume, consisting of the super-position of 5 consecutive ships, remained within the bounds of this region. We agree that for an individual ship one would expect a widening of the plume with distance. In previous analyses performed for this study we have quantified the core plume in these simulations where Na was outside 3 standard deviations of the background concentration, but this constrained the analysis to a rather narrow region of highest concentration around the emission line. No further insight was obtained from these results and they were therefore omitted in the manuscript. We therefore felt, that it was more insightfull to distinguish in the analysis between a region where CCN concentrations were elevated and a region where no increase in CCN was detected.

P12l7-10: This paragraph is not clear.

The paragraph has been rephrased for clarity.

P12l28: Observational studies showed ship tracks closing open cells (e.g., Goren and Rosenfled 2012 where at least part of the open cells seems to be deep, based on the cells spatial scale; Christensen and Stephens 2012). While simulations does not show a reverse transition, observational evidences should be provided as well.

The "large open cells" in Goren & Rosenfeld (2012) are estimated to be around 20-25km, which is roughly half the size of the cells simulated here. Furthermore, observational evidence (Durkee et al 2000b, Toll et al 2017, Chen et al 2015, Christensen & Stephens 2012) suggests that ship tracks in boundary layers deeper than 1km are extremely unlikely. Therefore, we would not expect to simulate a track-like structure in these simulations. However, it seems that a similar process occurs in deep boundary layers, where the large open cells partially fill in, but the filaments never stretch across the entire cell, which would then allow it to recover (given sufficient mixing generated through cloud-top cooling to overcome sub-cloud stability). The Goren and Rosenfeld reference was added to the manuscript (P12L33).

P16l17: Consider changing "such tracks" to "linear shaped tracks", and to add that they are rare in deep boundary layers in comparison to shallow boundary layers (reference is needed).

Rephrasing was done and references from the introduction repeated here.

P16l28-29: How do the fractional percentage calculated? From which table?

The paragraph was rephrased slightly such that now all numerical results are merely summarised here without reference, but are referenced to to corresponding figures and tables in section 3.

P16l26 the → the.

Done.

P16l27 annular → annual.

Done.

Figure 5: In the caption the boundary layer depth for each of the simulations should be provided (i.e., shallower in Wang et al 2011).

The information was added.

Figure 6: In order to cover also the night time in Figure 6d, consider replacing (or adding) cloud optical thickness with cloud albedo?

As only day-time values in Aall and Acld are considered throughout the paper, the night-time values are not shown here for consistency. Although they are diagnosed, we ommitted them in the averaging process, as they have no physical meaning but affect the temporal mean due to the diurnal cycle (clouds are optically thicker at night).

Supplementary: Caption 3: What is "ship_open"?

This typo was removed.

Caption 6: remove "and" in line 3. Caption is not clear. X axis label is not consistent.

This was changed.

Summary
10/15/2018 1:22:47 PM

Differences exist between documents.

**New Document:**
Manuscript_revised_submission
22 pages (11.53 MB)
10/15/2018 1:22:40 PM
Used to display results.

**Old Document:**
ACPD_submission_manuscript_July11th_2018
21 pages (12.07 MB)
10/15/2018 1:22:40 PM

Get started: first change is on page 1.

No pages were deleted

**How to read this report**

Highlight indicates a change.
Deleted indicates deleted content.
▲ indicates pages were changed.
⟷ indicates pages were moved.

[revised manuscript text omitted]